# Evaluating Trustworthiness in Reactive Web Architectures: A Structured Framework and Comparative Analysis

## Abstract

Reactive web architectures form the foundation of interactive web applications, but issues concerning trustworthiness arise with respect to maintaining application state, handling errors, and developer control. This paper describes a structured assessment framework that can be used to evaluate trustworthiness in different reactive approaches, whether it is based on signals, observables, or a combination of both. The notion of trustworthiness is abstracted using six different aspects, namely predictability, transparency, debuggability, failure isolation, user experience consistency, and developer oversight, and is demonstrated using scenario-based assessments of standardized interaction scenarios like asynchronous updates and concurrent user interactions, yielding a comparative study that reveals different architectural trade-offs, like better transparency in observable-based systems or better locality in signal-based systems.

## CCS Concepts

• **Software and its engineering** → **Software verification and validation**.

## Keywords

reactive programming, trustworthiness, evaluation framework, software architecture, scenario-based evaluation, web systems

## 1 Introduction

Modern web applications have increasingly used reactive web architecture to enable responsive user interfaces, propagating changes in state automatically upon every user interaction or asynchronous flow of data [2]. Applications in the lines of social media platforms, e-commerce systems, and data-intensive dashboards all rely heavily on real-time updates and handling complex dependency logic [7].

As these systems are increasingly employed in high-stakes and user-facing contexts, trustworthiness has emerged as a critical concern [1]. In addition to functional correctness, reactive web architectures need to support predictable state transitions, transparent data flows, effective debugging, and robust handling of partial failures [6]. Opaque mechanisms for updating state, unintentional state propagation, and limited observability can undermine developer confidence and user trust as systems grow in complexity.

Despite the considerable interest in reactive programming models and architectural patterns, the current state of evaluation of reactive web architectures remains piecemeal [2], [4]. Many previous works focus on performance-related metrics like latency or throughput, and few provide guidance regarding how architectural decisions affect more general properties of trustworthiness [4]. The lack of organized frameworks for evaluation inhibits the principled comparison between paradigms and limits transferring best practices across application domains [5].

The proposed paper will address this gap by suggesting a structured evaluation framework in assessing the trustworthiness of reactive web architectures. We decompose the notion of trustworthiness into six operational dimensions: predictability, transparency, debuggability, failure isolation, UX consistency, and human oversight. Further, we introduce a scenario-based methodology for comparative assessment. By applying this framework to signal-based, observable-based, and hybrid reactive architectures, actionable insights are obtained on architectural trade-offs and cross-domain implications for responsible web system design.

This work presents a framework-based qualitative evaluation of signal-based, observable-based, and hybrid reactive architectures, focusing on architectural trade-offs rather than empirical performance benchmarking.

### 1.1 Contributions

The contributions of this paper are as follows:

- We are proposing a multi-dimensional assessment framework that can operationalize trustworthiness in the context of reactive web architectures using six well-defined dimensions.
- We introduce a scenario-based comparative methodology that allows the reproducible assessment of signal-based, observable-based, and hybrid reactive paradigms.
- We derive key cross-domain insights that inform responsible and trustworthy design practices for modern web systems.

## 2 Related Work

Reactive programming has also received extensive attention in the literature as a paradigm for handling asynchronous data flows as well as dynamic changes in the state in modern software systems. A thorough survey on reactive programming is offered by Bainomugisha et al. [2], which classifies different methods of reactive programming on the basis of data flow dependencies, event handling strategies, as well as semantics, while also pointing out the challenges in each approach in regard to compositionality, expressiveness, as well as execution order. Later studies have tackled the manner in which principles of reactive programming are implemented in software engineering. Salvaneschi et al. [7] describe an overview of reactive programming principles and how these are important in structuring asynchronous computations and managing the evolution of state in complex applications. This discussion demonstrates that there are clear advantages in declarative data flow abstractions and how it becomes hard to deal with the control flow structures when developing applications.

Reactive architectural design patterns are also studied in domain-specific environments. Curasma and Estrella [4] present a survey on reactive software architecture in Internet-of-Things environments, specifically concerning real-time data, coordination, performance, etc. Their work might seem to concentrate on the Internet-of-Things domain, yet some common architect patterns in their study, such as event coordination, are common to reactive Web design patterns.

In addition to reactivity itself, trustworthiness has also been an area of long-standing interest within dependable and secure computing. Avizienis et al. [1] describe a basic categorization framework for dependability; they define properties such as reliability, availability, safety, and maintainability. These properties form the basic foundation for any forms of system trustworthiness beyond correctness.

The role of developer knowledge and debugging is also a very essential aspect in designing a trustworthy system. Ko et al. [6] examine developer mental models in program behavior using "why" and "why-not" queries, highlighting that reduced observability and complex state interactions present substantial barriers to effective subsequent debugging efforts, a consideration that is especially prominent in reactive systems, in which asynchronous processing and hidden dependencies can make it difficult to identify cause-and-effect relationships.

Recent work has explored signal-first architectural approaches for reactive systems, emphasizing fine-grained state propagation and improved developer reasoning about reactivity [3].

Finally, structured methods for the assessment of software architecture have been proposed to enable well-founded decision-making. Ref. [5] describe the Architecture Tradeoff Analysis Method (ATAM) to assess architecture decisions against orthogonal quality attributes such as performance, modifiability, or reliability. However, ATAM offers a general methodology to assess architecture designs but fails to address the specific characteristics of the Reactive Web architecture paradigm at hand, such as the propagation of fine-grained state transitions.

Beyond reactive programming and architectural evaluation, related work in program comprehension motivates the human-centric dimensions used in this paper. In particular, prior studies emphasize that observability, traceability, and mental-model alignment strongly shape developer effectiveness when diagnosing faults and understanding system behavior [8]. These insights support treating debuggability and human oversight as first-class concerns when evaluating reactive web architectures.

In summary, while current literature provides a strong starting point for reactive programming, dependability, debugging, and architectural evaluation, these topics have not been significantly integrated for reactive web architectures yet. Current surveys have concentrated on generalized principles or performance-related issues [2, 4], while architectural evaluation approaches [5] have not addressed trustworthiness issues in reactive systems. Thus, there is a need for a comprehensive evaluation approach to tackle trustworthiness in reactive web architectures.

## 3 Trustworthiness Dimensions and Evaluation Framework

Software trustworthiness can be considered to involve not only functional correctness properties, which make a system work correctly, but other properties like observability, controllability, and fault-tolerant properties, which make a system act in a predictable way, be understandable for developers, and work well even if its surroundings change [1]. It is even more important in reactive Web applications because their programming models include asynchronous computations, implicit control flow, and fine-grained control flow dependencies, making control flow properties hard to predict.

To facilitate the evaluation of trustworthiness in the context of reactive web architectures in a principled manner, we introduce a formal evaluation framework using six dimensions of trustworthiness. Our evaluation framework draws upon well-established principles of dependable software engineering and evaluation of software architectures [1, 5] to provide a formal means of comparison in the context of reactive architectural styles.

Today's reactive systems showcase various architectural patterns and failure behaviors that are hard to assess with single point measures. We therefore introduce a structured evaluation framework that logically relates reactive architectural styles with realistic system scenarios and trustworthiness dimensions. Figure 1 depicts the end-to-end flow of the proposed framework.

*Reactive Architectures.* We consider three representative reactive paradigms: signal-based architectures, observable-based architectures, and hybrid architectures. Signal-based architectures model application state as a dependency graph of signals with deterministic propagation. Observable-based architectures rely on event streams and subscriptions to propagate changes asynchronously. Hybrid architectures combine these approaches by using signals for core state representation while employing observable streams for side effects, asynchronous coordination, or external event handling (e.g., signal-driven state with observable-based effects).

In practice, this pattern appears in modern reactive web stacks where fine-grained signal graphs manage UI state, while observable streams orchestrate asynchronous workflows such as network requests, user event batching, or cross-component coordination.

*Evaluation Scenarios.* Architectures are tested for these representative stress conditions, such as asynchronous updates, concurrent interactions, partial failures, and cross-device latency, that can be incurred when the architecture is scaled and deployed.

*Trustworthiness Dimensions.* The system behavior is gauged through six parameters associated with trust levels: predictability, transparency, debuggability, failure isolation, UX consistency, and human oversight. All these parameters of system behavior are associated both with system reliability and human factors.

*Comparative Insights.* Through the correlation of architecture patterns and the behaviors exhibited based on various levels of trustworthiness, it is now possible to have comparative analysis capabilities within the architecture.

### 3.1 Trustworthiness Dimensions

**Predictability** can be described as the level at which programmers can foresee when and why changes to a system's state happen. Predictability in a reactive system architecture is based on the clarity of resolved dependencies and the predictability of updates. System architectures that have low side effects and allow localized reasoning make it easier for programmers to have a clear mental model of system behavior [6].

**Transparency**: This reflects the visibility of the data flow and dependency relations between components. In transparent systems,

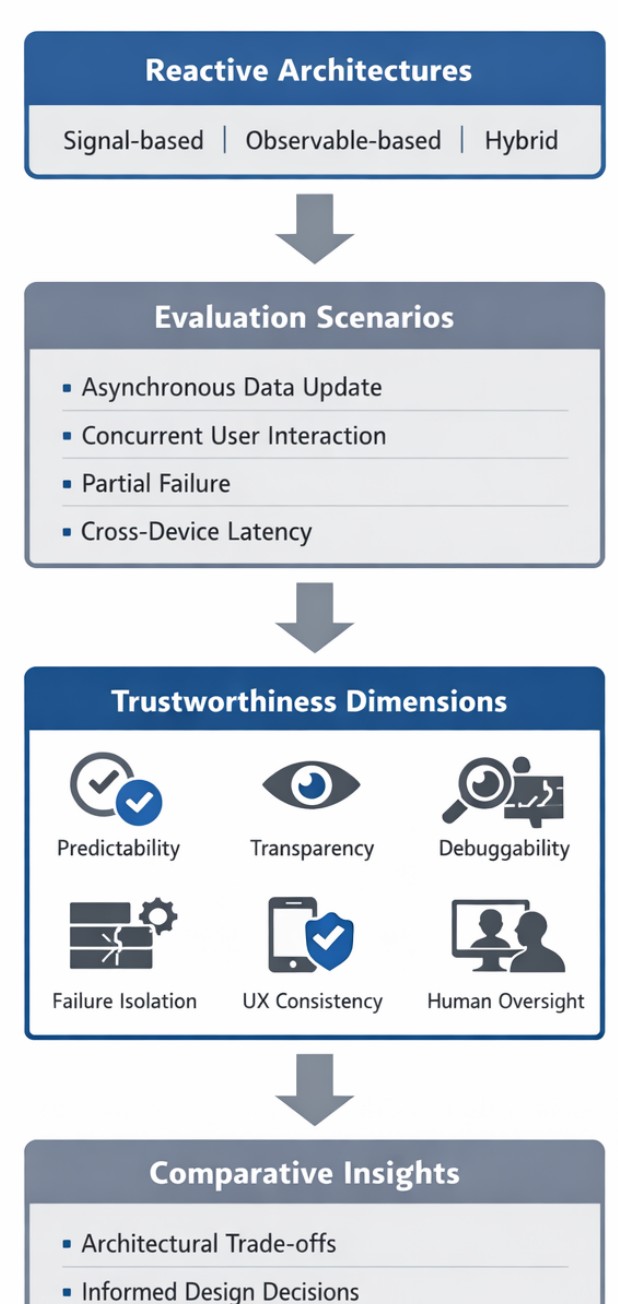

**Figure 1: Framework for evaluating reactive architectures across trustworthiness dimensions under realistic system scenarios.**

it becomes possible for developers to monitor the path of the inputs and events in the system and their effects on the system's observable behavior. Previous research on the topic of reactive programming states that the representation of the data flow can improve the

transparency properties of the system but might also add more complexity as the system evolves [7].

**Debuggability** systematically measures the ability to detect, isolate, and fix issues. Debugging in reactive system remains complex because of the asynchronous processing and non-linear control flow. Software debugging studies show a system enabling the debugging of execution traces and localized state analysis facilitates debugging process efficiency and successfully fixes issues [6].

**Failure Isolation** deals with the capacity for architectural containment of errors, inhibiting the spread of localized errors throughout unrelated system components. Within the realm of reactive systems, failure isolation is directly affected by the structure of state dependencies and error handling semantic definitions. Unfavorable isolation results in situations such as cascading failures that spread inconsistencies throughout the whole system, thus affecting system trustworthiness [1]. From a security perspective, effective failure isolation also limits the propagation of malicious or malformed events within reactive execution flows.

**UX Consistency** specifies the level of consistency in user experience despite network latency and other conditions such as simultaneous interactions. Reactive systems may exhibit UX inconsistencies even when application logic is correct if update ordering becomes nondeterministic under concurrency.

**Human Oversight**: This pertains to the ability of programmers to monitor, control, and change certain operations by systems, if needed. Trustworthy systems must enable adequate control by humans, especially in complex systems or safety applications. The design architecture may reduce oversight in control flow or coupled automated responses, which may increase risks in operations [1].

To distinguish worst-case stress behavior from typical usage, we report both scenario-specific assessments and an aggregated summary. Scenario-specific results (e.g., concurrent interactions) highlight conditions where architectural limitations are most pronounced, while the aggregated table summarizes expected behavior across a representative set of scenarios.

$$Score(A, d) = \sum_{s \in S} w_s \cdot score(A, d, s) \tag{1}$$

Here, $A$ denotes an architecture, $d$ a trustworthiness dimension, $s$ an evaluation scenario, and $w_s$ a scenario weight.

This explains why Hybrid can score lower in the concurrent interaction scenario (Table 2) while still scoring higher overall (Table 1) when considering the full set of representative scenarios.

## 3.2 Evaluation Framework Overview

The proposed evaluation framework applies a set of trustworthiness criteria outlined above to the standardized interaction scenarios to evaluate the reactive web architectures. Instead of empirical benchmarks and implementations, the framework uses structured reasoning and comparative analysis. This follows the approach used to perform scenario-based architectural system evaluations as presented in [5]. The framework assesses how effectively the system, represented by each architectural paradigm being evaluated, provides for each dimension of trustworthiness when being exposed to an identical scenario. In turn, this provides a comparative view of the trustworthiness characteristics of the paradigms, showing

benefits, drawbacks, and trade-offs in an impartial fashion that does not favor any paradigm in the comparative analysis to be presented in the subsequent section.

The following tables distinguish between scenario-specific stress evaluations and aggregated trustworthiness assessments to provide complementary perspectives on architectural behavior.

Table 1 highlights systematic differences across architectural paradigms. Signal-based systems exhibit strong predictability and debuggability due to explicit dependency graphs, while observable-based ones prioritize transparency and failure isolation through subscription models. Hybrids balance most dimensions but may introduce overhead in real-world scaling.

In particular, concurrency stress scenarios can expose semantic interaction costs in Hybrid designs, even when Hybrid performs strongly in non-concurrent or well-structured interaction patterns.

*3.2.1  Qualitative Scoring Criteria.* To reduce ambiguity and improve reproducibility, qualitative scores used throughout the evaluation are assigned according to the following criteria:

- **High**: The architectural property is enforced by design and consistently holds across all evaluated scenarios.
- **Medium**: The property holds under typical conditions but may degrade under stress scenarios such as high concurrency or partial failures.
- **Low**: The property is not consistently enforced and relies on external mechanisms or developer discipline to maintain.

These criteria are applied uniformly across architectures and scenarios to ensure consistent interpretation of the comparative assessments.

## 4  Scenario-Based Comparative Evaluation

The proposed trustworthiness assessment approach is then demonstrated on three main web-reactive architecture styles: signal-based architectures that focus on fine and detailed state updates [2]; observable-based architectures that are based on subscription-based models of change distribution [7]; and finally, the third category that consists of the combination of both styles that look to enhance flexibility [4]. The assessment is qualitative and based on properties and reasoning related to scenarios that are common in other approaches related to architectural tradeoff studies [5].

### 4.1  Evaluation Scenarios

The scoring examples are to be architecture-independent and cover typical issues found in reactive web systems:

- **Asynchronous Data Update**: A web query updates common application data, causing related computations and updates to be triggered.
- **Concurrent User Interaction**: There are simultaneous events triggered by the user, and they compete to change the state.
- **Partial Failure**: A fault occurs in one of the components in the state propagation that tests the ability to contain the error.
- **Cross-Device Latency**: Update states are received in a reordered manner because of differences in network latency.

These scenarios are designed to reflect common sources of complexity in reactive systems, such as asynchrony, concurrency, partial failure, and nondeterminism in update order.

### 4.2  Assessment and Results

In each of the scenarios, we evaluate the level to which each of the architectural paradigms supports the six trustworthiness dimensions introduced in section 3. Table 2 illustrates the results obtained in the *Concurrent User Interaction* scenario. These results follow a similar trend across the remaining evaluation scenarios, with concurrency-driven interactions most strongly exposing architectural trade-offs in predictability and failure isolation.

For Hybrid architectures, the coexistence of signal-based propagation and observable-based event streams introduces semantic heterogeneity. Under concurrent interaction scenarios, this heterogeneity can complicate reasoning about update ordering and failure containment, which explains the lower predictability and failure isolation scores observed in Table 2.

Signal-based systems show high predictability based on their dependencies on the state and update functions, while being fully deterministic. Observable-based systems offer better transparency and debug properties, since flow dependencies are described in detail using data flow chains [6, 7]. A hybrid approach has better flexibility, while in some cases, it may show lower predictability and weaker isolation in case of failures [5].

In signal-based approaches, failures can often be contained within localized dependency regions or effect boundaries, which improves isolation under many conditions. However, isolation depends on how effects are implemented and how error boundaries are defined, especially under concurrent updates.

Observable-based approaches typically expose explicit error channels and recovery operators (e.g., catch/retry patterns), which can support containment but may also propagate failures broadly if stream composition crosses unrelated concerns.

Hybrid approaches must reconcile failure semantics across signal graphs and observable streams. Under concurrency, this boundary can complicate containment and recovery, which helps explain lower failure-isolation scores in stress scenarios.

UX consistency follows related trends: architectures that ensure deterministic update ordering and stable state snapshots under concurrency are less likely to surface inconsistent intermediate UI states.

### 4.3  Illustrative Case Study: Applying the Framework to a Concrete Signal-First Architecture

To make the proposed framework more concrete and reduce perceived subjectivity, we provide an illustrative case study demonstrating how the trustworthiness dimensions and qualitative scoring criteria can be applied to a specific reactive architecture artifact.

*Artifact.* We use the Signal-First reactive architecture described in [3] as the concrete reference system. Signal-First models application reactivity as a directed acyclic dependency graph with explicit primitives (Signals, Computed, Effects) and schedules effects only

**Table 1:** Overall trustworthiness assessment across reactive architectures. Ratings (High / Medium / Low) follow the qualitative scoring criteria defined in Section 3.2.1. This table presents an aggregated view across representative interaction scenarios, reflecting typical system usage rather than worst-case behavior.

| Architecture Type | Predictability | Transparency | Debuggability | Failure Isolation | UX Consistency | Human Oversight |
|---|---|---|---|---|---|---|
| Signal-based | High | Medium | High | Medium | High | Medium |
| Observable-based | Medium | High | Medium | High | Medium | Medium |
| Hybrid | Medium | High | Medium | Medium | High | Medium |

**Table 2:** Scenario-specific trustworthiness assessment under concurrent and stress interaction patterns. Ratings are assigned using the qualitative criteria in Section 3.2.1 and emphasize worst-case or diagnostic scenarios (e.g., concurrent updates and asynchronous interactions) where architectural limitations are most pronounced.

| Dimension | Signal-based | Observable-based | Hybrid |
|---|---|---|---|
| Predictability | High | Medium | Low |
| Transparency | Medium | High | Medium |
| Debuggability | Medium | High | Low |
| Failure Isolation | High | Medium | Low |
| UX Consistency | Medium | High | Medium |
| Human Oversight | Medium | Medium | High |

study is intended as an illustrative application of the framework rather than a validation of architectural superiority, and is used to demonstrate how qualitative assessments can be grounded in a real system artifact.

**Table 3:** External empirical anchor from the Signal-First architecture artifact. Reported values are used solely to ground runtime and scalability discussion and do not replace the qualitative scoring procedure.

| Reported indicator | Value |
|---|---|
| Mean update latency (stress) | 71.8 ms |
| Memory usage (stress baseline) | < 1 MB vs. 8.4 MB |
| Stress-condition speedup | 5.7× |

after graph stabilization, yielding deterministic propagation and consistent state snapshots.

*Scenario instantiation.* We instantiate the paper's *Concurrent User Interaction* scenario (Section 4.1) as overlapping state updates driven by rapid UI events and asynchronous effect completions. In Signal-First, concurrent updates are reconciled through ordered propagation over the dependency graph and deferred effect execution after stabilization.

*Dimension-by-dimension scoring (illustrative).* Using the qualitative criteria in Section 3.2.1, we illustrate the mapping from architectural mechanisms to trustworthiness scores.

**Predictability:** Signal-First emphasizes deterministic propagation over the dependency graph and stable snapshots before effects run, supporting a *High* predictability assessment.

**Failure Isolation:** Separating pure computations (Computed) from impure side effects (Effects) localizes error surfaces and supports containment; recovery mechanisms remain implementation-specific, motivating a *Medium-to-High* score depending on the application's error-boundary strategy.

**Debuggability / Human Oversight:** Explicit reactive primitives and stabilized execution phases enable clearer mental models and improved traceability of state propagation compared to implicit subscription-based chains.

*Lightweight empirical anchor.* Although this paper does not introduce new benchmarking, the Signal-First artifact reports stress-condition measurements that serve as an external empirical anchor for scalability-related discussion, such as reduced update latency and lower memory usage at large reactive graph sizes. These values are used only to ground scalability and runtime-cost discussion and do not redefine the qualitative scoring procedure. This case

## 4.4 Synthesis of Insights

Regardless of the scenarios, a set of trade-offs still hold. Whereas signal-based architectures prioritize locality and error containment, but possibly lack global visibility of dependencies [1], observable-based paradigms encourage transparency and debuggability in dependency-intensive applications and are therefore appropriate in the context of complex interactive systems [6], while hybrid architectures represent a compromise with the potential of trustworthiness problems related to semantic heterogeneity and increased architectural complexity [5]. On the whole, the assessment points out that trustworthiness in responsive web architectures is determined by trade-offs within the architectures rather than by either paradigm. The developed framework facilitates the discussion of trade-offs to make informed architectural decisions.

## 5 Discussion and Implications

By using the scenario-based evaluation, it is shown that no architectural paradigm reaction optimizes all the dimensions of trustworthiness. Rather, the different dimensions of trustworthiness in the reaction architectural paradigm work in web applications by means of architectural trade-off, especially in localized control and global observability.

## 5.1 Tooling and Ecosystem Effects

Although the proposed framework primarily addresses architectural properties, there are scenarios where the tooling offered by the developers can make some aspects of trustworthiness dimensions more relevant than others. Tooling doesn't remove architectural trade-offs but provides a degree of relief from them.

For instance, debugging aids that provide information regarding dependency graphs, execution traces, or state transition history

may enhance transparency or debuggability in systems that are not automatically transparent or debuggable.

Likewise, static analyzers as well as code lints can assist in spotting any unwanted graphs of dependencies or side effects. However, such tooling is implemented on top of the architecture. Architectures which are more transparent in how the state or data flow can more easily be made the focus of tool-oriented mitigation techniques. Architectures which have less transparent control flow could potentially reduce the efficacy of tool-oriented mitigation. Consequently, it is important for tooling to be considered a complement rather than a substitute for architectural trustworthiness.

**Table 4: Illustrative mapping between trustworthiness dimensions and common classes of developer tooling.**

| Dimension | Example Tooling Support |
|---|---|
| Predictability | Invariant checks, dependency analyzers |
| Transparency | Dependency graph visualizers, trace inspectors |
| Debuggability | Execution tracing, time-travel debugging |
| Failure Isolation | Fault injection tools, error boundary analysis |
| Human Oversight | Monitoring dashboards, intervention hooks |

## 5.2 Runtime Costs, State Management, and Recovery Mechanisms

Beyond architectural reasoning, trustworthiness is influenced by runtime costs, state management strategies, and recovery mechanisms that emerge during real-world deployment. While these factors do not redefine architectural guarantees, they shape how trustworthiness dimensions manifest at scale.

*Runtime costs.* Observable-based architectures may incur additional runtime overhead due to long-lived subscription chains, event buffering, and intermediate stream objects, which can increase memory usage and garbage collection pressure as application complexity grows. Signal-based architectures typically emphasize direct dependency propagation, which can reduce intermediate allocations but may introduce overhead when large dependency graphs require frequent invalidation and recomputation. Hybrid architectures combine both cost profiles and therefore require careful coordination to prevent compounded overhead at semantic boundaries.

*State management libraries.* State management libraries play a mediating role between architectural paradigms and developer-facing behavior. Libraries such as RxJS enhance observable-based systems by offering structured operators and error-handling primitives, improving transparency and debuggability while potentially increasing runtime overhead. Signal-oriented libraries and frameworks integrate state propagation more directly, supporting stronger predictability and failure containment at the cost of reduced global visibility in large graphs.

*Recovery mechanisms.* Failure isolation in reactive architectures depends not only on error containment but also on recovery strategies such as retries, fallback state restoration, or compensating updates. Observable-based systems often support explicit recovery

through retry and catch operators, whereas signal-based systems rely on error boundaries and effect-level guards to prevent cascading failures. Hybrid architectures must reconcile these approaches, making recovery behavior more complex and emphasizing the importance of explicit architectural governance.

**Table 5: Illustrative relationship between reactive architectures, runtime costs, and recovery characteristics.**

| Architecture | Runtime cost tendency | Typical recovery style |
|---|---|---|
| Signal-based | Graph recomputation overhead | Error boundaries, guarded effects |
| Observable-based | Subscription and buffering overhead | Retry, catch, stream restart |
| Hybrid | Combined overhead profiles | Mixed recovery semantics |

## 5.3 Granularity Versus Observability

Signal-driven architectures are designed to focus on state management at a much finer granularity, supporting effective isolation and predictability for failures. Unfortunately, while such a level of granularity might hide global dependency relationships, making it harder to debug paths through which system state changes, at scale, designers will need to use discipline to compensate for lack of visibility.

In contrast, the architecture based on observables implies a higher degree of expressiveness for dependencies. This is well aligned with previous work that has noted that knowledge of execution paths improves the understanding of software [6, 7]. The trade-off is that this incurs higher cognitive complexity because of the need for understanding asynchronous streams.

**Table 6: Examples of observable proxies and tooling support for trustworthiness dimensions.**

| Dimension | Observable Proxy | Typical Tooling Class |
|---|---|---|
| Predictability | Update ordering consistency | Static analysis, invariant checks |
| Transparency | Dependency visibility | Dependency graphs, tracing tools |
| Debuggability | Time to isolate faults | Execution traces, step debugging |
| Failure Isolation | Error containment scope | Fault injection, recovery monitors |
| Human Oversight | Intervention visibility | Monitoring dashboards, logs |

*5.3.1 Security as a Cross-Cutting Trustworthiness Concern.* Security concerns in reactive web systems are closely intertwined with architectural trustworthiness rather than isolated properties. Vulnerabilities such as unintended event injection, improper propagation of untrusted inputs, or manipulation of reactive flows can directly undermine predictability, failure isolation, transparency, and human oversight.

For example, reactive architectures that allow uncontrolled propagation of externally triggered events may increase the risk of cascading failures or inconsistent state updates. Similarly, limited

visibility into reactive execution paths can hinder detection and mitigation of security-relevant anomalies.

Rather than introducing security as a separate dimension, the proposed framework evaluates security-relevant concerns through their impact on existing trustworthiness dimensions. This perspective aligns with architectural reasoning practices in dependable system design, where security, safety, and reliability are evaluated in relation to containment, observability, and control mechanisms.

## 5.4 Scalability and Complexity Effects

With increased size and complexity of reactive systems, dimensions of trustworthiness could deteriorate in varied manners depending on the architectural paradigm. Moreover, an increase in the number of state dependencies, interaction paths, and asynchronous coordination points can potentially increase the complexity of reasoning.

Scalability issues could arise in signal-processing architectures when there are large graphs of dependencies, in which case global transparency could be compromised even when local predictability is high. Observable-based architectures may see the cognitive overhead increase due to the number of events and subscriptions being handled. This might result in issues related to the predictability and debuggability of the application.

Hybrid architectures could also have their difficulties compounded because of interaction effects from more than one reactive semantics. The problem of failure isolation could also become more difficult. These serve to point out that trustworthiness is not fixed; it is scalable. The framework, consequently, enables reasoning about architectural decisions both independently and in terms of their behavior as applications evolve.

## 5.5 Architectural trade-offs in High-Stakes Contexts

For applications where human evaluation and predictability are important, architectural consistency is an important factor for trustworthiness too. The hybrid architecture intends to capitalize on the strengths of more than one reactive paradigm, while it appears that combining reactive semantic models will bring in ambiguity in execution order and error propagation, since semantic variability might enhance complexity levels for reasoning and undermine isolated failure, thus reflecting architectural evaluation concerns on complexity [5].

In relation to dependability, more explicit architectures concerning control flow and error propagation are more amenable to reasoning about system behavior in exceptional circumstances [1]. It is not being suggested that a hybrid strategy is not a good fit, merely that it is a strategy for which trust governance is essential.

## 5.6 Implications for UX Consistency

Consistency in UX in reactive web applications is related to predictability in the ordering and propagation of updates. Architectures that can easily enable reasoning in a deterministic manner in the face of concurrent inputs are better equipped to handle consistent UX. More observable-centric models of programming can offer better abstractions for dealing with concurrency, but they also require greater programmer diligence to ensure they don't create unexpected interactions. More straightforward reactive models may help to curb the overhead in simpler use cases, but they may not scale to complex UX easily.

## 5.7 Implications for Responsible Design

This framework treats trustworthiness as an architecturally significant concern that must be addressed explicitly during system design rather than emerging incidentally from performance optimization.

Instead of prescribing one particular architecture and prescribing what aspects of trustworthiness must be pursued, the framework helps make well-informed decisions about which aspects of trustworthiness are given importance. This also corresponds with traditional notions within software engineering that dependability, transparency, and human factors must be incorporated within architecture design [1, 5].

Although the evaluation focuses on reactive web architectures, the framework is applicable to other event-driven and reactive systems, including data-intensive dashboards, monitoring interfaces, and human-in-the-loop systems.

## 5.8 Limitations

The current work takes a qualitative viewpoint based on the architecture level and does not incorporate framework-specific optimisation, tool ecosystems, or empirical performance characteristics. Although this abstraction facilitates generality, it might be useful in extending the framework to incorporate empirical validation analysis in order to improve the estimation of trustworthiness in the future. Future work could extend the framework to more explicitly operationalize security-related threats and adversarial behaviors within reactive architectures. In particular, large-scale systems may exhibit non-linear degradation of certain trustworthiness dimensions, which motivates future empirical and tool-assisted studies.

## 6 Conclusion and Future Work

This paper presented a structured framework for evaluating trustworthiness in reactive web architectures. By decomposing trustworthiness into six operational dimensions—predictability, transparency, debuggability, failure isolation, UX consistency, and human oversight—we provided a systematic lens for analyzing how architectural choices influence trustworthy system behavior. Applying the framework through scenario-based comparative evaluation, including an illustrative case study grounded in a concrete signal-first architecture (Section 4.3), we highlighted recurring trade-offs across signal-based, observable-based, and hybrid reactive paradigms.

The evaluation demonstrates that trustworthiness is not an inherent property of any single reactive architecture but rather an emergent outcome of design decisions and contextual priorities. Signal-based architectures emphasize locality and fault containment, observable-based approaches favor transparency and debuggability, and hybrid architectures balance expressiveness with increased reasoning complexity. The proposed framework enables these trade-offs to be articulated explicitly, supporting informed architectural decision-making without relying on empirical benchmarks.

From a broader perspective, this work aligns with the goals of multifaceted evaluation emphasized by the TIME workshop. By focusing on architectural reasoning and scenario-based assessment, the framework supports responsible design practices across application domains where reactive systems are increasingly deployed.

Future work may extend this framework in several directions. Empirical studies could validate and refine the qualitative assessments presented here by examining real-world systems and developer experiences. Tool-assisted analyses could automate parts of the evaluation process, enabling scalable assessment of trustworthiness properties. Finally, the framework could be adapted to emerging paradigms, such as reactive systems integrated with AI-driven components, to further explore trustworthiness challenges in evolving web architectures. By emphasizing transparency, debuggability, and human oversight, the framework supports broader Responsible AI and trustworthy web system design goals.

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
