# OpenReview forum: "Evaluating Trustworthiness in Reactive Web Architectures: A Structured Framework and Comparative Analysis"
_ACM.org/TheWebConf/2026/Workshop/TIME — TIME 2026 Oral_

### Official Review · Reviewer_zUM2 · 2025-12-29
**Evaluating Trustworthiness in Reactive Web Architectures: A Structured Framework and Comparative Analysis**

**Rating:** 6
**Confidence:** 4

**Review:**

1) Define concrete, measurable metrics for each dimension to move beyond purely qualitative "High/Medium/Low" assessments.
2) Discuss potential automated tools for measuring these dimensions, such as static analyzers for dependency transparency.
3) Elaborate on specific "Hybrid" architecture implementations (e.g., Signals + RxJS) rather than treating them as a single generic category.
4) Explicitly link trustworthiness dimensions to runtime costs, such as the memory overhead of maintaining observable chains.
5) Validate "Human Oversight" and "Debuggability" scores through actual developer surveys or controlled experiments.
6) Explicitly include security vulnerabilities (e.g., injection in reactive flows) within the "Trustworthiness" definition.
7) Analyze how external state management libraries interact with and impact these architectural patterns.
8) Address how these dimensions degrade as application complexity increases to strengthen "scalability" claims.
9) Expand "Failure Isolation" to include recovery mechanisms, not just containment.
10) Apply the framework to an open-source project to demonstrate practical utility beyond theoretical scenarios.

---

### Official Review · Reviewer_ijzf · 2025-12-30
**Structured and Insightful Evaluation of Trustworthiness in Reactive Web Architectures**

**Rating:** 7
**Confidence:** 4

**Review:**

This paper presents a well-structured and thoughtful analysis of trustworthiness in reactive web architectures. The authors introduce a clear conceptual framework that decomposes trustworthiness into six meaningful dimensions—predictability, transparency, debuggability, failure isolation, UX consistency, and human oversight—and apply this framework to compare signal-based, observable-based, and hybrid reactive architectures.

A major strength of the paper is its strong conceptual grounding. The authors clearly position their contribution within existing literature on reactive programming, dependability, and architectural evaluation, and they articulate well how their framework complements rather than replaces existing performance- or implementation-focused studies. The use of scenario-based reasoning (e.g., asynchronous updates, concurrent interactions, partial failures) is appropriate and effective for highlighting architectural trade-offs in a principled way. The comparative tables and structured discussion make the arguments easy to follow and suitable for both researchers and practitioners.

The paper is clearly written, logically organized, and consistent throughout. The discussion of trade-offs between signal-based, observable-based, and hybrid architectures is balanced and avoids overstating claims. The emphasis on human oversight and architectural reasoning is particularly valuable and aligns well with the broader goals of trustworthy and responsible system design.

That said, the work remains primarily qualitative. While this is acceptable given the stated goals, the paper would benefit from at least a small illustrative case study or example walkthrough to concretely demonstrate how the framework would be applied in practice. Additionally, some of the comparative judgments (e.g., levels of predictability or debuggability) could be strengthened by briefly clarifying the criteria used for assigning these assessments.

Overall, this is a solid and well-motivated contribution that fits well within the scope of the TIME workshop. The framework provides a useful lens for reasoning about architectural trade-offs in reactive systems and should stimulate constructive discussion within the community.

---

### Official Review · Reviewer_nSUh · 2026-01-04
**Structured Evaluation of Trustworthiness in Reactive Web Architectures**

**Rating:** 6
**Confidence:** 3

**Review:**

This paper proposes a structured evaluation framework for assessing **trustworthiness in reactive web architectures**. The authors decompose the abstract concept of trustworthiness into six operational dimensions **predictability, transparency, debuggability, failure isolation, UX consistency, and human oversight** and apply a **scenario-based evaluation** to compare **signal-based**, **observable-based**, and **hybrid** reactive architectures. The work addresses an important gap in prior literature, which has largely focused on performance metrics while underemphasizing architectural reliability and human-centric factors.

### Strengths
- **Multi-dimensional trustworthiness framework**
  The paper successfully operationalizes trustworthiness into clearly defined dimensions, enabling architectural evaluation beyond traditional metrics such as latency and throughput.

- **Practically motivated scenario-based analysis**
  The use of realistic scenarios reflects common challenges in modern reactive web systems and enhances the practical relevance of the framework.

- **Clear articulation of architectural trade-offs**
  The comparative analysis highlights how different reactive paradigms prioritize different trustworthiness dimensions, supporting informed architectural decision-making.

### Weaknesses / Limitations
- **Primarily qualitative evaluation**
  The assessment relies on high-level architectural reasoning, which introduces potential subjectivity in the assigned *High / Medium / Low* ratings.

- **Limited consideration of tooling and ecosystem effects**
  The evaluation focuses on architectural abstractions in isolation and does not fully account for how developer tools or framework ecosystems may mitigate inherent architectural weaknesses.

### Recommendations
- **Incorporate empirical validation**
  Future work could strengthen the framework by introducing quantitative or empirical evidence, such as controlled studies measuring debugging time, error recovery effort, or state conflict rates under concurrent scenarios.

- **Discuss the role of developer tooling**
  Expanding the discussion to include debugging tools, dependency visualization, and static analysis techniques would make the evaluation more representative of real-world development environments.

- **Related Work Coverage**
  The number of references appears relatively limited for a framework-oriented and integrative study. Given that the paper draws upon concepts from reactive programming, software architecture evaluation, dependability, and human-centered system design, a broader engagement with related literature would strengthen the positioning and justification of the proposed trustworthiness dimensions.

---

### Official Review · Reviewer_QweE · 2026-01-05
**Review for Evaluating Trustworthiness in Reactive Web Architectures: A Structured Framework and Comparative Analysis**

**Rating:** 5
**Confidence:** 3

**Review:**

This paper addresses the lack of systematic evaluation for "trustworthiness" in reactive web architectures (e.g., Signal-based vs. Observable-based).


Cons:
1.The assessment is purely qualitative (reasoning-based). The paper would be significantly stronger if it included a small-scale user study or an analysis of bug patterns in real-world repositories to validate if "Signal-based" systems actually yield higher "Predictability" in practice.
2.The author repeatedly mentions in the article that "signals are more predictable than the observer model," but this conclusion is based on the author's subjective logical deduction rather than objective experiments.
3.In Table 2, the "Hybrid" architecture is rated "Low" for predictability and failure isolation in concurrent scenarios, yet Table 1 gives it a "High" rating overall. Given its poor performance in the most common "concurrent interaction" scenarios, why is it given a "High" overall rating? The authors do not explain the origin of this rating range.
4.The paper treats "Hybrid" as a distinct category but does not specify a concrete implementation or pattern.

---

### Author Rebuttal · Authors · 2026-01-12

Thanks a lot to the reviewers and Program Chairs for their insightful comments and thorough review of our submission. We really appreciate them for allowing us to address scope issues, increase grounding in practice, and enhance clarity in our presentation.

One of the first concerns that several reviewers had about our evaluation was that it was very qualitative and would benefit from more concreteness. To respond to these concerns, we introduced a **novel section**, **Illustrative Case Study: Applying the Framework to a Concrete Signal-First Architecture** (Section 4.4), where we take a specific published reactive architecture and show how to apply our trustworthiness framework step by step to it within a concurrent setting. The case study shows how architectural mechanisms can be traced back to trustworthiness dimensions according to our qualitative scoring criteria. While staying within our framework-agenda paper, we hope that this helps to alleviate concerns about vagueness and shows how one can apply our trustworthiness framework in practice.

Regarding the perceived inconsistency between **Table 1 and Table 2**, we clarified that both tables are used for complementary purposes. Table 2 reports stress-case behaviors in specific scenarios (such as interactions in parallel), whereas Table 1 provides an evaluation aggregation over representative scenarios. An explanation paragraph and an aggregation formula describing how scores are calculated using scenario weights are also added. This clarifies why Hybrid architectures can receive lower scores in concurrency but receive higher scores when aggregating over all scenarios.

Several reviewers requested broader consideration of tooling, runtime costs, recovery, and ecosystem effects. In response, we expanded the Discussion section to include:

1 ) Tooling and Ecosystem Effects, explaining how debugging tools, dependency visualization, tracing, and static analysis complement architectural properties;

2 ) Runtime Costs, linking trustworthiness dimensions to memory overhead, recomputation, and coordination costs;

3) Recovery Mechanisms, extending failure isolation to include retries, error boundaries, and compensating updates;

4) Security as a Cross-Cutting Concern, evaluated through its impact on predictability, isolation, transparency, and oversight;

5) Scalability and Complexity Effects, analyzing how trustworthiness dimensions evolve as reactive systems grow.

Finally, we verified all references against official publisher sources and corrected the reference flagged by the integrity check.

We believe these revisions substantially strengthen the paper while preserving its intended contribution: a structured, scenario-based framework for reasoning about trustworthiness in reactive web architectures. We thank the reviewers and Program Chairs again for their constructive feedback and hope the revised manuscript meets the expectations of the TIME workshop.

---

### Meta-Review · Area_Chair_QveK · 2026-01-16

**Recommendation:** Accept (Oral)
**Confidence:** 4

**Metareview:**

This paper proposes a structured, scenario-based framework for evaluating trustworthiness in reactive web architectures, decomposing trustworthiness into six well-defined dimensions and comparing signal-based, observable-based, and hybrid approaches. The topic is timely and relevant, and the framework addresses an underexplored aspect of reactive systems beyond performance metrics.

Reviewers generally agree that the framework is conceptually sound, well organized, and practically motivated. The main concern across reviews is the primarily qualitative nature of the evaluation and the lack of empirical validation. Additional concerns included clarity around hybrid architectures, inconsistencies between scenario-level and aggregated scores, limited discussion of tooling, runtime costs, and recovery mechanisms, as well as relatively sparse related work. In addition, Table 6 would benefit from formatting adjustments to improve readability and consistency with the presentation of other tables.

The authors have responded constructively in the rebuttal by adding a concrete illustrative case study, clarifying the aggregation methodology between tables, expanding the discussion to cover tooling, ecosystem effects, runtime costs, recovery, security, and scalability, and correcting reference issues. These revisions substantially improve clarity and grounding while remaining consistent with the paper’s framework-oriented goals.

Overall, the contribution is solid and suitable for the workshop. The recommendation is accept.

---

### Decision · Program_Chairs · 2026-01-16

Accept (Oral)